# Citations Network Analysis of Vision and Sport

**DOI:** 10.3390/ijerph17207574

**Published:** 2020-10-18

**Authors:** Henrique Nascimento, Clara Martinez-Perez, Cristina Alvarez-Peregrina, Miguel Ángel Sánchez-Tena

**Affiliations:** 1ISEC Lisboa, Instituto de Educação e Ciência de Lisboa, 1750-179 Lisboa, Portugal; henrique.nascimento@iseclisboa.pt; 2Faculty of Biomedical and Health Science, Universidad Europea de Madrid, 28670 Madrid, Spain; cristina.alvarez@universidadeuropea.es; 3Faculty of Sport Sciences, Universidad Europea de Madrid, 28670 Madrid, Spain; miguelangel.sanchez@universidadeuropea.es

**Keywords:** sport, vision, performance

## Abstract

*Background*: Sports vision is a relatively new specialty, which has attracted particular interest in recent years from trainers and athletes, who are looking at ways of improving their visual skills to attain better performance on the field of play. The objective of this study was to use citation networks to analyze the relationships between the different publications and authors, as well as to identify the different areas of research and determine the most cited publication. *Methods*: The search for publications was carried out in the Web of Science database, using the terms “sport”, “vision”, and “eye” for the period between 1911 and August 2020. The publication analysis was performed using the Citation Network Explorer and CiteSpace software. *Results*: In total, 635 publications and 801 citations were found across the network, with 2019 being the year with the highest number of publications. The most cited publication was published in 2002 by Williams et al. By using the clustering functionality, four groups covering the different research areas in this field were found: ocular lesion, visual training methods and efficiency, visual fixation training, and concussions. *Conclusions*: The citation network offers an objective and comprehensive analysis of the main papers on sports vision.

## 1. Introduction

Over the years, the field of ophthalmology has placed great emphasis on the idea of achieving “normal vision” (20/20). Nevertheless, there is much more to perfect vision than having normal vision. While the term “eyesight” refers to the clarity of the image in the retina, the term “vision” has a broader meaning, which encompasses the mental process of deriving meaning from what is seen. Therefore, vision is the result of visual pathway integrity, visual efficiency, and visual information processing [1,2].

Sports vision is a relatively new specialty in the field of optometry, and its objective is to improve and preserve visual function to increase sports performance. Its beginning dates back to the 18th century when the good eye began to occlude in amblyopic patients; however, concerns about athletes’ vision did not emerge until the 20th century, when an optometrist began to advise a group of athletes in the United States [3]. In the 1960s, visual examinations were performed on a baseball team in the United States, and, in the 1970s, optometrist services began to be offered routinely to athletes [4]. The first time that a series of visual tests was performed was in the 1984 Olympics, which were held in Los Angeles; in the 2004 Olympics, which were held in Athens, visual tests were conducted on a large number of the athletes. In 1988, the European Academy of Sports Vision was created in Rome with the aim of training specialist technicians in sports vision [4]. These days, sports vision is considered as a very important discipline for the preparation of athletes in the United States.

Many studies have demonstrated that vision plays a vital role in good sports performance [1,2,5]. Most of the athletes and trainers that participated in these studies demonstrated that sports performance requires a wide range of perceptive, technique, psychological, and physical skills. Each sport requires a combination of visual skills, which are essential to ensure adequate sporting performance. By training these specific visual skills, athletes will exhibit better skills and effectiveness in the field of play [6]. Likewise, over the last few decades, perception has been acknowledged as a key aspect of the field of play [7,8,9,10]. The study by Spera et al. [11] evaluated and compared balance with both open and closed eyes and the strength of the lower extremities in sighted and visually impaired athletes. The results of the research showed that postural stability was different as a function of the evaluation with the eyes closed and open. Furthermore, the comparison between blind and sighted judo athletes highlighted greater difficulties with closed eyes for sighted athletes than for blind ones. In this way, they showed that vision loss significantly affects performance, especially if athletes do not have a congenital visual deficit, but rather progressively lose vision.

Burris et al. [12] concluded that visual–motor skills play an important role in sports performance; therefore, it has been suggested that sensorimotor skills could be a useful tool when examining players. Another investigation observed that elite athletes have better cognitive skills, with volleyball players demonstrating highly flexible attention and superior executive control [13,14,15].

Ciućmański et al. [16] demonstrated that, in terms of peripheral vision, depth perception, and the ability to visually track a moving object, footballers had better results than their nonathletic peers. This is since training focusing on developing visual perceptive abilities increased the levels of these abilities and consequently the efficiency of an athlete’s perception.

Basic elements of sports vision include visual reaction time and peripheral vision [17]. Both factors significantly affect the athlete’s perceptive skills, although they have fundamentally different precedents. Peripheral vision is influenced by the general functions of the human visual system. On the other hand, visual reaction time is related to information and the cognitive processes that control and regulate movement, and these are affected by the central nervous system functions and the muscular effects. Motor reaction time is the time between the signal and the completion of an action; therefore, it has both sensory and motor characteristics [18]. In this way, handball players receive most of their information through their vision, with the player having to pay attention to more than two different stimuli, for instance, an unmarked teammate or a close opponent. That is to say, optimal central–peripheral simultaneity is vital as this allows the player to take in all of the visual information about the object that their vision is focused on, as well as all that is happening around them, without having to make any ocular movement [19,20].

Sports vision training makes use of stimuli in optometric exercises such as videos, images, or stroboscopic interruptions of vision. It is based on the idea that improving visual skills through oculomotor exercises can be associated with motor actions, thereby resulting in an improved sports performance [21,22,23]. The study performed by Abernethy et al. [24] used generic stimuli (alphanumeric symbols, shapes, patterns, and colors) that were presented in painted graphics or objects. Participants had to answer with a simple ocular adjustment which was combined with simple motor actions such as pointing or touching objectives. In another study, sports training (university football, basketball, or throwing and catching exercises) was assessed using Nike Vapor Strobe glasses, comparing the results to those of athletes who had trained with training eyewear. Participants wearing the Strobe glasses had better results in terms of central visual field motion sensitivity and transient attention abilities than the control group. However, no differences were found in terms of their peripheral motion sensitivity or in terms of their multiple-object tracking [23]. In 2003, García Manso et al. [25] concluded that “vision constitutes a hugely important tool in sports practice and, therefore, visual education must occupy a special part in the athletes’ training, primarily when the tasks to be performed are open”.

Citation network analysis is used to search scientific literature on a specific subject. In other words, citations can help us find other publications that may be of interest, to demonstrate, both qualitatively and quantitatively, the relationships that exist between articles and authors through the creation of groups [26]. Furthermore, it is possible to quantify the most cited publications in each group and, likewise, we can study the development of a research area or focus the literature search on a specific subject [26,27,28,29].

Therefore, taking into consideration the increasing number of publications on sports vision, this study aimed to identify the different research areas and determine the most frequently cited publication. Likewise, it aimed to analyze the relationships between the publications and the different research groups by using the CitNetExplorer software (Ness Jan van Eck and Ludo Waltman, Centre for Science and Technology Studies (CWTS), Leiden University, Leiden, The Netherlands), which examines the development of the scientific literature in a specific research field.

## 2. Materials and Methods

### 2.1. Database

The search of publications was carried out in the Web of Science (WOS) database, using the following search terms: “sport”, “vision”, and “eye”. These terms were selected as the study objective because they are the most common terms in all of the research fields.

As the search results had articles in common, the boolean NOT and AND operators were used, and the truncation symbol * was used to search for the singular and plural form of the terms. The first search used the terms (“sport* vision”), the second search used the terms (“sport*” AND “eye” NOT “sport* vision”), and the third search used the terms (“sport*” AND “vision” NOT “sport* vision”). Additionally, the search field was classified by topics, and the results were limited by abstract, title, and keywords. The selected timeframe was from 1911 to August 2020.

Web of Science also makes it possible to add references to your library while conducting bibliographic searches directly in external databases or library catalogs.

Several citation indexes were used in our study, namely, the Social Sciences Citation Index, the Science Citation Index Expanded, and the Emerging Sources Citation Index.

Likewise, given how certain authors and institutions cite works may vary, the CiteSpace software (Chaomei Chen, College of Computing and Informatics, Drexel University, PA, USA) was also used in order to standardize the data. The publications were searched and downloaded on 27 August 2020.

### 2.2. Data Analysis

The Citation Network Explorer software was used to analyze the publications, as it is a tool that allows the researcher to analyze and visualize the citation networks of scientific publications and even download these directly from Web of Science. Managing citation networks obtained in this way makes it possible for the researcher seeking to analyze a certain subject to use a citation network comprising several million publications and related citations as the starting point for a deeper analysis that will eventually yield a smaller subnetwork of 100 publications.

A quantitative analysis of the most-mentioned publications within a specific timeframe was conducted using the citation score attribute. As such, not only were the internal connections within the Web of Science database quantified, but also any external connections, meaning that other databases were considered [30].

The Citnetexplorer provides several techniques for analyzing publications citation networks. The clustering functionality is achieved using the formula developed by Van Eck in 2012 [30].
(1)$$V\left(c_{1\;},\ldots ,c_{n}\right)=\underset{i<j}{\sum }\delta \;\left(c_{i},c_{j}\right)\left(s_{ij}-\gamma \right).$$

This functionality was used to assign a group to each publication. As a result, the most related publications tended to be found in the same group as a function of citation networks [30].

Finally, the core publications were analyzed using the identifying core publications functionality. This functionality serves to identify the publications that are considered to be the core of a citation network, that is to say, publications with a minimum number of connections with other core publications, thereby allowing irrelevant publications to be eliminated. The researchers established the number of connections in the knowledge that a higher value of this parameter denoted a lower number of core publications [30]. In this way, this study considered publications that presented four or more citations in the citation network.

Additionally, the drilling down functionality was used, as it enables the researcher to conduct a deeper analysis of each of the groups at different levels.

On the other hand, CiteSpace software (5.6.R2) was used to perform scientometric analysis. This software, developed by Chen Chaomei, is based on Java language and it comprises five basic theoretical aspects: Kuhn’s model of scientific revolutions, Price’s scientific frontier theory, the organization of ideas, the best information foraging theory of scientific communication, and the theory of discrete and reorganized knowledge units [31,32]. A specific assessment can also be conducted within the scientometric analysis process by using certain parameter indicators. Physicist Jorge Hirsch (University of California, USA) proposed the H index, a mixed quantitative index that can provide an assessment of the level and amount of academic output produced by a certain researcher or academic institution. This index, computed by evaluating the number of citations for given papers within a journal, is used as an indicator showing that *h* of the *N* published papers have been cited at least *h* times. It is also used to quantify the productivity and impact of a group of researches that belong to a department, university, or country. It should be noted that, if the software yields a value of 1, this does not constitute a professional answer [33]. The parameter indicator “degree” is used to show the number of co-occurrences between authors, institutions, or countries in the knowledge graph; consequently, more communication and cooperation between them would result in a higher degree value. Likewise, the importance of nodes in the research cooperation network and the continuity of institutional research over time can also be measured by using the intermediary centrality and the half-life indicators, respectively [31].

### 2.3. Ethical Approval

This study was approved by the ethics committee of the General Directorate for Research and Development (DGID) of Instituto Superior de Educação e Ciências (ISEC) Lisbon, Portugal. The ethical approval number is 01/27052020.

## 3. Results

The first articles about sports vision were published in 1911; thus, the selected time interval was from 1911 to August 2020. In total, 635 publications and 801 citations networks were found in the search that was conducted in WOS (Figure 1). Of all the publications, 73.75% were articles, 8.85% were proceedings, 5.90% were reviews, 4.13% were congress and conference summaries, 3.24% were letters, and 2.06% were book chapters.

The number of publications about sports vision has increased significantly since 2011 (1911–2010: 34.96% of the publications; 2011–2020: 65.04% of the publications). The year 2019 had the largest number of publications, accounting for 68 publications and six citations networks (Figure 2).

Table 1 shows the 20 most cited publications in this citation network. The most cited article was the article by Williams et al. [34], which was published in November 2002, with a citation index of 28. This study analyzed the relationship among the “quiet eye” (final fixation on the target before the initiation of movement), expertise, and task complexity in a near and a far aiming task in 24 billiards players (12 professional players and 12 less skilled players). In order to do so, two experiments were established, which were based on establishing the different visual fixation time during the preparation phase of the action. They found that shorter quiet eye periods resulted in poorer performance, irrespective of participant skill level. Therefore, the authors argued that quiet eye duration represents a critical period for movement programming in the aiming response.

The 20 most cited articles were analyzed. Five of these discussed ocular lesions associated with sport [36,47,48,49,52], four discussed the use of visual training for improving sports skills, 10 discussed training visual fixation skills (“quiet eye”) [24,34,35,37,38,39,40,41,42,43,44,45,46,51], and the final article was about the significance of training of oculomotor movements in subjects with sports-related concussion [50].

### 3.1. Description of the Publications

The research on sports vision is multidisciplinary. The fields of sport science (17.09%) and ophthalmology (13.25%) (Table 2) are particularly worth mentioning. Table 3 shows the 10 journals with the largest number of publications.

As shown in Table 4, the authors with the largest number of publications on sports vision were Mann (4.27%), Balcer (2.56%), and Galetta (2.56%).

The United States (20.94%), England (12.82%), and China (8.97%) were the countries with the highest publication rate (Table 5).

Additionally, the most used keywords were “implantation” (325 publications), “contrast sensitivity” (285 publications), and “performance” (231 publications). Table 6 shows the 30 most used keywords from the most significant publications.

### 3.2. Clustering Function

The clustering function identified four groups, all of which contained a significant number of articles (Figure 3). Table 7 shows the information on the citation networks for the four main groups, listed by size from the largest to the smallest.

In group 1, 108 publications and 276 citations were found throughout the network. The most cited publication was the article by MacEwen et al. [36], which was published in 1987 in the *British Journal of Ophthalmology*. In this study, a survey was performed with all patients presenting with sport-related ocular lesions and a total of 246 patients presented with this type of injury during an 18-month period. Football was responsible for 110 (44.7%), rugby for 24 (9.8%), squash for 19 (7.7%), badminton for 16 (6.5%), and ski for nine (3.7%), while 68 (27.6%) were caused by other sports. In total, 46 patients (18.7%) required inpatient care and 200 (81.3%) were treated as outpatients, of whom 104 required at least one follow-up appointment (42.3% of the total). The authors concluded that, with increasing time available for leisure activities, there has been a parallel increase in sport associated eye traumas. The studies in this group dealt with ocular lesions associated with sport, as well as their prevalence (Figure 4).

In group 2, 80 publications and 202 citations were found throughout the network. The most cited publication was the article by Stine et al. [39], which was published in 1982 in the *Journal of the American Optometric Association*. This article addressed two studies that demonstrated that athletes have better visual skills than non-athletes and, in turn, that better athletes have better visual abilities than poorer athletes. They also confirmed that visual skills are trainable and transferable to athletic performance, meaning that players present a larger extent of visual field, larger fields of recognition, larger motion perception fields, lower amounts of heterophoria at near and far ranges, more consistent simultaneous vision, more accurate depth perception, better dynamic vision acuity, and better ocular motility, whereby all of these visual skills can be improved with appropriate visual training. The publications in this group addressed efficiency and the different methods of visual training with the purpose of improving the visual skills of athletes, leading to better on-field performance (Figure 5).

In group 3, 68 articles and 238 citations were found throughout the network. The most cited publication was the article by Williams et al. [34], which was published in 2002 in the *Journal of Motor Behavior*, which was also at the top of the list of the 20 most cited publications. The publications in this group addressed how visual fixation training (quiet eye) allows for better results to be achieved in the field (Figure 6).

In group 4, 28 publications and 40 citations were found throughout the network. The most cited publication was the article by Master et al. [49], which was published in 2016 in *Clinical Pediatrics*. This article looked to determine the prevalence of visual impairment after concussion in adolescents. In order to do so, 100 teenagers with a mean age of 14.5 years were examined, and it was found that 69% presented with one or more of the following vision diagnoses: accommodative disorders (51%), convergence insufficiency (49%), and saccadic dysfunction (29%). Therefore, they concluded that performing a visual examination is recommended and that these visual diagnoses must be taken into account when they return to the field of play. The publications in this group addressed the importance of evaluating oculomotor movements in patients with concussions, as this type of training can help athletes who present with or who have suffered concussions (Figure 7).

Table 8 shows a detailed description of the oldest and newest publications in the four main groups.

After using the drilling down functionality to analyze the relationships among the four main groups, no connections were found between them (Figure 8).

#### 3.2.1. Subclusters in Group 1

Four subgroups were found (Figure 9), three of which contained a significant number of publications (Table 9). The remaining group was relatively small with fewer than 12 publications and 15 citations networks.

#### 3.2.2. Subclusters in Group 2

Four subgroups (Figure 10) were found, of which three contained a significant number of publications (Table 10). The remaining group was relatively small with fewer than 12 publications and 13 citation networks.

### 3.3. Core Publications

In Figure 11, after the analysis was performed using the core publications functionality, 114 publications with four or more citations (16.9% of the publications) and 471 citations networks were obtained.

## 4. Discussion

The first publication about sports vision was published by Walker et al. [69] in 1911, just as concern for sports vision was beginning to emerge at the beginning of the 20th century. This article analyzed the importance of the dominant eye and how skills in shooting sports could be improved. However, publications on sports visions were still very limited until the mid-1990s; nonetheless, the number of papers being published was increasing and these were beginning to cover a range of topics [70]. Studies by Fullerton [71] and Winogrand [72], which were published in 1925 and 1942, respectively, demonstrated that athletes presented better visual skills than amateur athletes. Following this, a considerable number of studies analyzed the effect of visual training programs on visual skills and sports performance, with these studies modifying the experimental parameters such as stimuli or contexts [24,73]. Numerous studies have shown that visual perception skills and cognitive performance improve after a visual training program. In Clark et al.’s study [74], a team of baseball players was subjected to a visual skills training program that incorporated both traditional and technological methods. The results showed that the players’ batting averages and slugging percentage were better than those recorded in the previous season. In another study, Di Noto et al. [75] used the “rapid serial visual presentation” (RSVP) method to evaluate ocular movements and visual attention. In this study, the 20 subjects were divided equally into control and experimental groups, each of which performed a pre-training RSVP assessment where the target letters, to which subjects were asked to respond by pressing a spacebar, were serially and rapidly presented. The response time to target letters, the accuracy of correctly responding to target letters, and the correct identification of the target letters in each of the 12 sessions were measured. The experimental group then performed active eye exercises, while the control group performed a task that minimized eye movements for 18.5 min. A post-training RSVP assessment was performed by both groups, and the response time, accuracy, and letter identification were compared between and within the subject groups for both pre- and post-training. Subjects who performed eye exercises were more accurate in responding to target letters separated by one distractor and in letter identification in the post-training RSVP assessment, while the latency of responses was unchanged between and within groups. This suggests that eye exercises may prove useful in enhancing cognitive performance on tasks related to attention and memory over a very brief course of training, and RSVP may be a useful measure of this efficacy. This relates to the fact that athletes and their trainers are constantly looking for ways to improve their physical and mental skills, and, due to the demanding nature of sports, visual and motor skills are often the focal point of sports training programs.

At the same time, 2019 was identified as a key year, due to the considerable number of studies that were published that year and due to the progress that was made in the research into sports vision. Hausegger et al. [76] study that suggested that gaze anchoring is functional for optimizing the use of peripheral visual information was seen to be particularly relevant. This study predicted that the height of gaze anchoring on the opponent’s body would depend on the potential attacking locations that need to be monitored. To test this prediction, the authors compared high-level athletes in kung fu (Qwan Ki Do), who attack with their arms and legs, with Tae Kwon Do fighters, who attack mostly with their legs. As predicted, the results showed that Qwan Ki Do athletes anchor their gaze higher than Tae Kwon Do athletes do before and even during the first attack. Furthermore, gaze anchoring seems to depend on three factors: the particulars of the evolving situation, crucial cues, and specific visual costs (especially suppressed information pickup during saccades). Another relevant study published by Mashkovskiy et al. [77] analyzed how the degree of visual impairment influenced the outcomes in a judo team. The findings confirmed that blind athletes had fewer chances to win a judo fight given that the loss of vision functions affects movement coordination, balance, and emotional state, which are important for martial arts.

One journal with a particularly high number of publications about sports vision is *Optometry and Vision Science*, which occupies the 40th place in the ophthalmology category, and which boasts an impact factor of 1.46. Articles were published in 150 magazines in the ophthalmology topic category, which seems reasonable given that sports vision is a specialty of the field of optometry and ophthalmology. The journal with the highest impact factor was the *British Medical Journal*, 17.21. However, it is important to consider that, although the impact factor is a critical index of the journal’s importance, it is not an absolute measure index. The main difference between both indexes is that the latter is based on the impact of the research results, as well as the authors’ physical and intellectual contributions [78].

The countries with the largest number of published articles were the United States, England, and China. This is not surprising given that the first studies about sports vision were published in the United States and the sports vision section of the American Optometric Association, the oldest in the world, was founded in 1978. However, it is worth mentioning that the authors’ institution with the highest rate of publications was “Vrije Universiteit Amsterdam”, because, in recent years, both athletes and trainers have shown an increased interest in improving their visual skills, especially in competitive sports. The creation of the European Association of Sports Vision and the Sports Vision Association of the United Kingdom are also considered relevant. Equally, the upward trend in the numbers of publications from countries such as the United States or the United Kingdom has been linked to a combination of factors, for example, the fact that these are English-speaking countries or the possible connections that may exist between the different research groups within the scientific community [79,80].

Sports vision as a specialty in the ophthalmology and optometry field is constantly expanding. In recent years, research has commenced into the benefits of advanced visual training. In a study that looked to analyze the stroboscopic effects on anticipatory timing, the skills of athletes were compared before and after using the Bassin anticipation timer. The experimental group practiced with the Bassin timer wearing Nike Vapor Strobe glasses set to level 3 (100 ms clear/150 ms opaque), while the control group practiced with normal vision. The post-training assignments were administered immediately, 10 min, and 10 days after training. In comparison to the control group, the Strobe group was significantly more accurate immediately after training, and it was more likely to respond early than to respond late immediately after training and 10 min later [81]. Using low visual level tools, Deveau et al. [82], trained 19 players with the Ultimeyes app for 30 sessions of 25 min, while another 18 players did not undergo training as the control subjects. The binocular VA (visual acuity) and SC (contrast sensitivity) were tested before and after the training. Results showed improved VA and SC, and, after seven days, the VA was superior to the normal levels. In baseball, it led to a reduction in the number of strikeouts and an increase of 4–5 extra games won compared to previous years.

Vision allows muscles to respond to signals, that is, it provides information to the athlete about when and where current activity is occurring. Therefore, all athletes require good vision, in order to reduce head and body movements, analyze three-dimensional space, or clearly see an object in motion. However, depending on the sport, it is necessary that certain skills are more developed than others.

It is possible to create citation networks using the main databases such as Web of Science or Scopus. However, when conducting a systematic review of all the existing literature on a subject, their usefulness is limited, given that they do not offer a general overview of the connection between the citations of a group of publications. That is why the CitNetExplorer software was selected, as it allows the researcher to visualize, analyze, and explore the citation networks of scientific publications. As such, the CitNetExplorer offers a more detailed analysis when creating citation networks compared to other databases such as Web of Science or Scopus [30]. The main aim of this study was to analyze the existing literature on vision and sport. In order to do so, the Web of Science database was used. The Web of Science database is one of the most comprehensive databases, as its search range goes back to the year 1900. Nevertheless, it is important to take into consideration the fact that the Web of Science (WOS) only accepts international journals once they have undergone a rigorous selection process.

Therefore, once the existing bibliography was downloaded from the WOS, the CitNetExplorer and CiteSpace software allowed us to collect and analyze every available piece of literature on sports vision to date. Furthermore, by analyzing the citation networks, it was possible to obtain the connection between the fields of study and the different research groups. The clustering function was used to collect the results, and the publications were then grouped according to the relationships between the citations. The drilling down function was used to examine the existing bibliography for each group, and the core publication function was used to show the main publications, that is to say, those with a minimum number of citations. These functions, therefore, made it possible for a complete study and analysis of the research on the field of study to be conducted.

Regarding the limitations of this study, publications where title, abstract, or keywords did not contain the search terms may not have been considered. Furthermore, if we compare this study with systemic reviews, explicit and prespecified methods were used to identify, evaluate, and synthesize all the available evidence related to a clinical question. Where appropriate, systematic reviews may include a meta-analysis, that is, a statistical combination of results from two or more separate studies. Some systematic reviews compare only two interventions, in which a conventional peer-to-peer meta-analysis can be performed, while others examine the comparative effectiveness of many or all available interventions for a given condition. Therefore, the analysis of citation networks allows a broader analysis of the bibliography that exists on a given topic.

In perceptive–cognitive training, the NeuroTracker system was investigated on a football field. The precision of passes, dribbling, and shooting was compared during small-sided games in university-level football players. The experimental group was formed by nine players who trained with the NeuroTracker system for 10 sessions, while the active control group was formed by seven players who trained for 10 sessions with three-dimensional (3D) football videos, and the passive control group was formed by seven players who did not receive any training. The results indicated that there were improvements in the players’ passing accuracy, but no improvements were noted in terms of dribbling and shooting between the pre- and post-sessions in the players in the NeuroTracker group, compared with those in the control groups. Moreover, the result was correlated with the players’ subjective decision-making accuracy, rated after pre- and post-sessions through a visual analogue scale questionnaire. These results indicate that the training exercise with the NeuroTracker protocol could selectively improve dynamic performance skills which are important for the sports performance [83].

However, although there is an increasing interest in the training of visual skills to improve sports performance, it is not clear whether or not visual training will improve performance on the field of play. This is related to the lack of scientific evidence supporting the efficacy of vision training on sports performance, as a result of a focus on the methodology, which results in a lack of validity of the training methods [70].

In the coming years, future research will focus on developing a diverse range of training programs to continue training the visual skills that are most relevant in sport in order to improve performance on the field of play. Furthermore, sports vision techniques can also be used to evaluate or rehabilitate sports-related concussions. Additionally, the expansion of the sports vision discipline could be used as a platform to develop a closer interprofessional connection between ophthalmology and optometry. Offering eye care to athletes is a field in which the possibility of greater synergy, mutual respect, and an exchange of knowledge between professionals is possible. In addition, interaction with other related health disciplines is possible and may lead to significant discoveries.

Consequently, the number of studies being published on sports vision is on the rise, given the need for more scientific evidence to demonstrate the positive effect that visual training programs have on sports performance. Regarding citation network studies, these are more numerous, given that it is the only analysis method providing a global overview of the different research fields within a specific topic. Furthermore, the CitNetExplorer and Citespace software allow analyzing all existing research on a specific topic through detailed studies. This could change how studies in different research areas are conducted.

## 5. Conclusions

In conclusion, this research offered an exhaustive and objective analysis of the main articles on sports vision. In this study, it was possible to visualize, analyze, and explore the most cited articles and citation networks existing to date using the Web of Science database and the Citation Network Explorer software.

Sports vision is a relatively new specialty; therefore, more scientific evidence is required in order to confirm the benefits of visual training in the sports field. All athletes must be aware of the importance of the visual system and the impact that it can have on sports performance. In some countries such as the United States, visual training is performed on all athletes on a routine basis; however, in other countries, it remains a very much unknown specialty.

The extant bibliography shows a wide variety of tools and options in the field of visual training, from basic optometry materials to special advanced systems for sports optometry. There are infinite ways in which these visual training tools can be combined to improve results by transferring skills from the training room to the playing field. Therefore, performing multisensorial and integrated visual training is ideal as it simultaneously works the different aspects. Furthermore, performing training exercises in the field is recommended.

## Figures and Tables

**Figure 1 ijerph-17-07574-f001:**
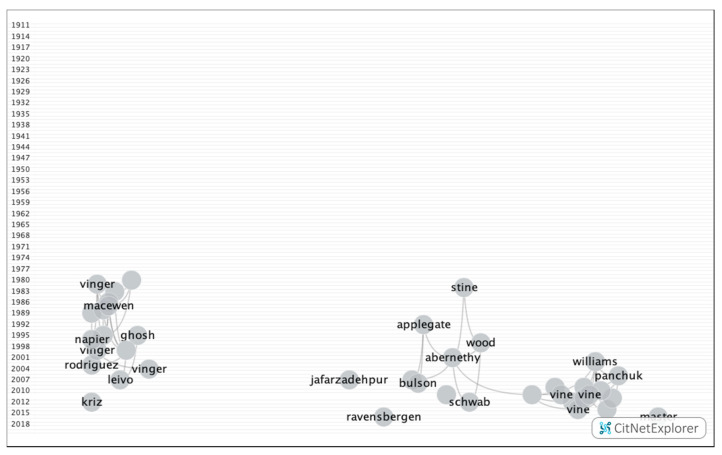
Citation networks on sports vision.

**Figure 2 ijerph-17-07574-f002:**
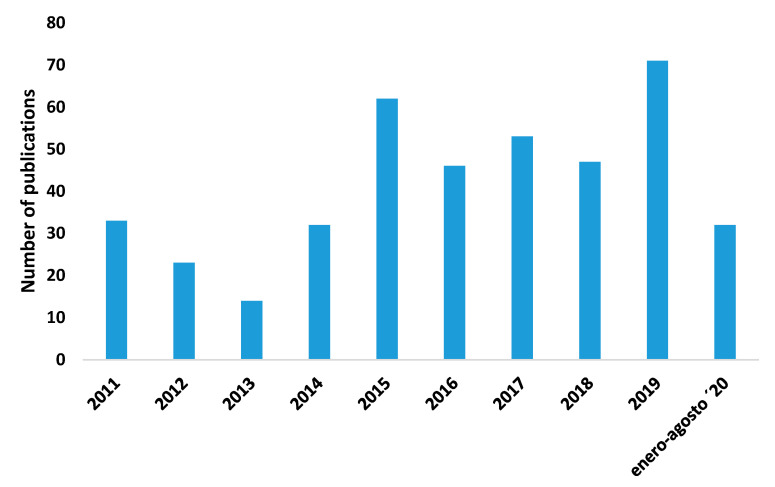
Number of publications per year.

**Figure 3 ijerph-17-07574-f003:**
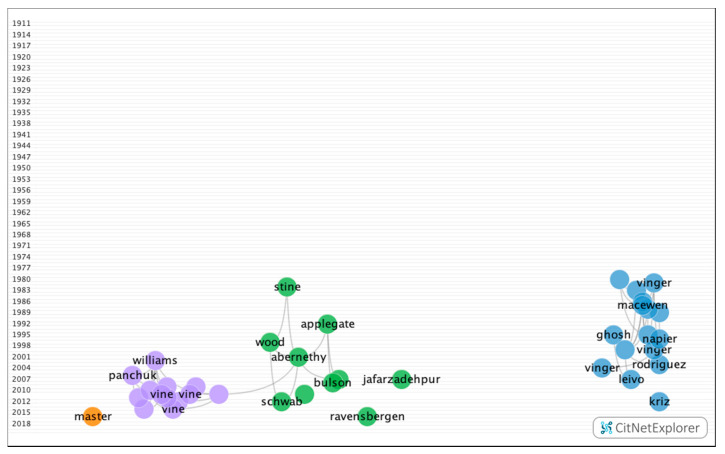
Clustering function in the citations network of sports vision.

**Figure 4 ijerph-17-07574-f004:**
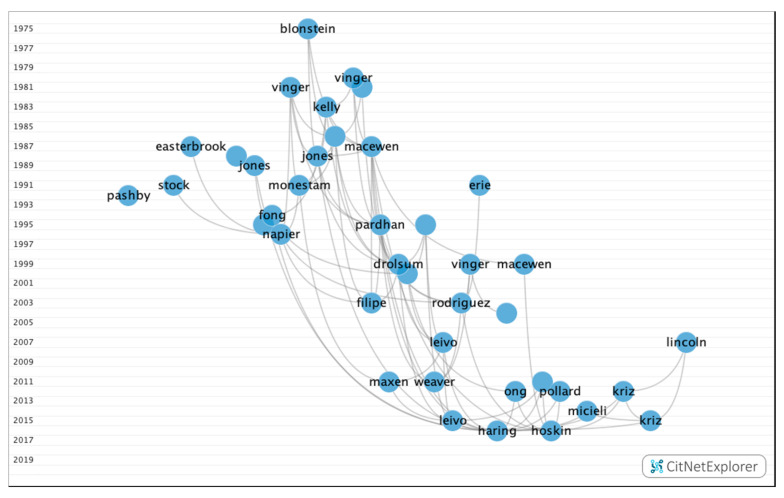
Citation network in group 1.

**Figure 5 ijerph-17-07574-f005:**
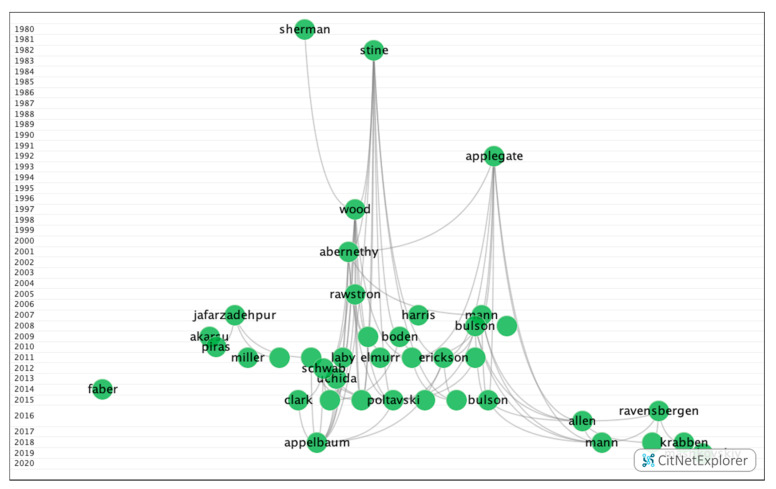
Citation network in group 2.

**Figure 6 ijerph-17-07574-f006:**
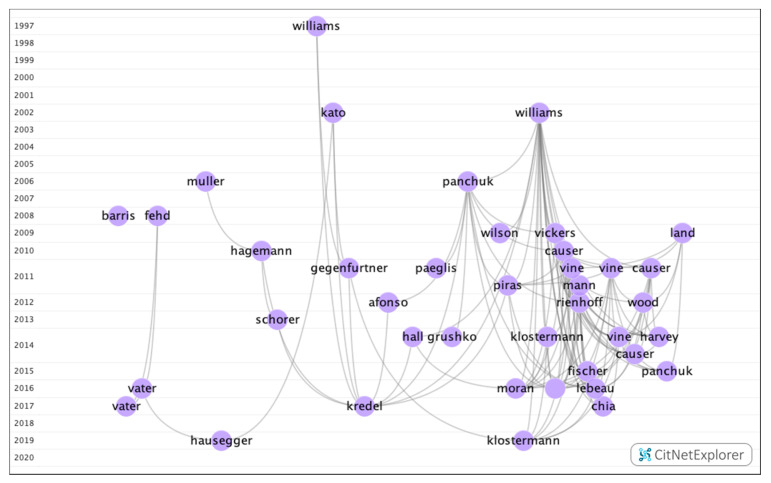
Citation network in group 3.

**Figure 7 ijerph-17-07574-f007:**
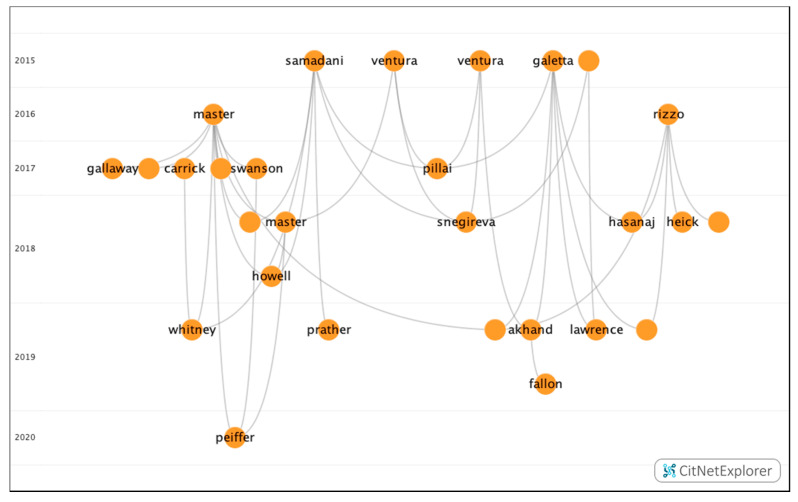
Citation network in group 4.

**Figure 8 ijerph-17-07574-f008:**
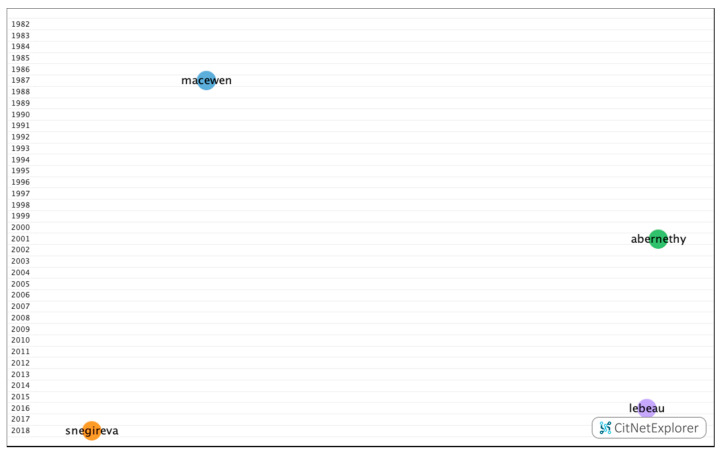
Connection between the four main groups.

**Figure 9 ijerph-17-07574-f009:**
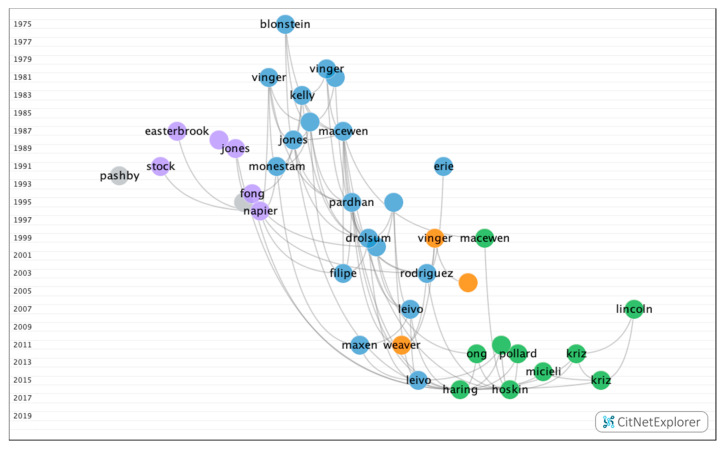
Citation network of subclusters in group 1.

**Figure 10 ijerph-17-07574-f010:**
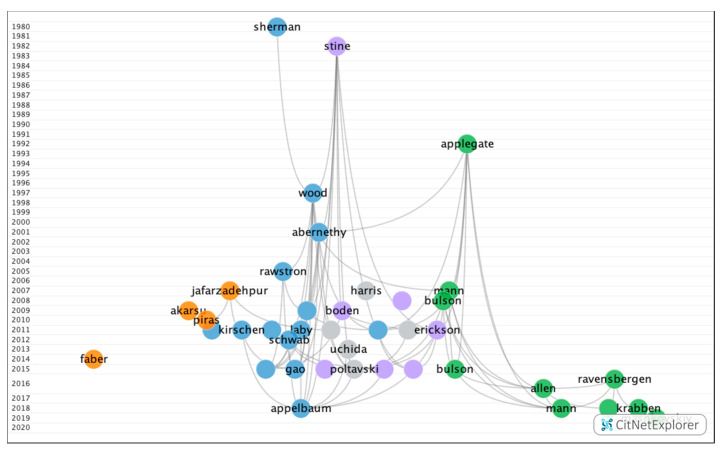
Citation network from the subclusters in group 2.

**Figure 11 ijerph-17-07574-f011:**
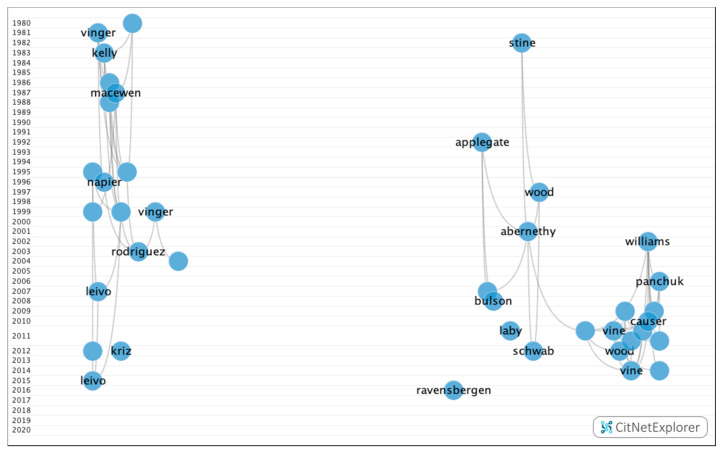
Core publications in the citation network about multifocal IOLs.

**Table 1 ijerph-17-07574-t001:** Description of the 20 most cited publications on sports vision.

Author	Title	Journal	Year	Total Number Citations	Citation Rate	*h*-Index
**Williams et al. [34]**	Quiet eye duration, expertise, and task complexity in near and far aiming tasks	*J. Mot. Behav*. **2002**, *34*, 197–207	2002	28	1.55	1
**Vine et al. [35]**	Quiet eye training facilitates competitive putting performance in elite golfers	*Front Psychol*. **2011**, *8*, 8	2011	21	2.33	1
**MacEwen et al. [36]**	Sport-associated eye injury: a casualty department survey	*Br. J. Ophthalmol*. **1987**, *71*, 701–705	1987	18	0.54	1
**Vine et al. [37]**	The influence of quiet eye training and pressure on attention and visuo-motor control	*Acta Psychol. (Amst)*. **2011**, *136*, 340–346.	2011	17	1.89	1
**Causer et al. [38]**	Quiet eye duration and gun motion in elite shotgun shooting	*Med. Sci. Sports Exerc*. **2010**, *42*, 1599–1608	2010	17	1.70	1
**Abernethy et al. [24]**	Do generalized visual training programs for sport really work? An experimental investigation	*J. Sports Sci.* **2001**, *19*, 203–22	2001	16	0.84	1
**Stine et al. [39]**	Vision and sports: a review of the literature	*J. Am. Optom. Assoc*. **1982**, *53*, 627–633.	1982	16	0.42	1
**Mann et al. [40]**	Quiet eye and the Bereitschaftspotential: visuomotor mechanisms of expert motor performance	*Cogn. Process*. **2011**, *12*, 223–234	2011	15	1.67	1
**Vickers et al. [41]**	Advances in coupling perception and action: the quiet eye as a bidirectional link between gaze, attention, and action	*Prog. Brain Res*. **2009**, 174, 279–288	2009	14	1.27	1
**Panchuk et al. [42]**	Gaze behaviors of goaltenders under spatial–temporal constraints	*Hum. Mov. Sci*. **2006**, *25*, 733–752	2006	14	1.00	1
**Wood et al. [43]**	An assessment of the efficacy of sports vision training programs	*Optom. Vis. Sci*. **1997**, *74*, 646–659.	1997	13	0.56	1
**Vine et al. [44]**	Quiet eye training: the acquisition, refinement and resilient performance of targeting skills	*Eur. J. Sport Sci*. **2014**, *14 (Suppl. 1)*, S235–S242	2014	12	2.00	1
**Causer et al. [45]**	Quiet eye training in a visuomotor control task	*Med. Sci. Sports Exerc*. **2011**, *43*, 1042–1049.	2011	12	1.33	1
**Vinger et al. [46]**	Sports eye injuries a preventable disease	*Ophthalmology* **1981**, *88*, 108–113.	1981	12	0.31	1
**Napier et al. [47]**	Eye injuries in athletics and recreation	*Surv. Ophthalmol. Nov.-Dec* **1996**, *41*, 229–444.	1996	11	0.46	1
**Gregory et al. [48]**	Sussex Eye Hospital sports injuries	*Br. J. Ophthalmol.* **1986**, *70*, 748–750	1986	11	0.32	1
**Master et al. [49]**	Vision diagnoses are common after concussion in adolescents	*Clin. Pediatr. (Phila)* **2016**, *55*, 260–267	2016	10	2.50	1
**Schwab et al. [50]**	The impact of a sports vision training program in youth field hockey players	*J. Sports Sci. Med*. **2012**, 11, 624–631.	2012	10	1.25	1
**Vinger et al. [51]**	Sports-related eye injury. A preventable problem	*Surv. Ophthalmol. Jul.-Aug.***1980**, *25*, 47–51	1980	10	0.25	1
**Klostermann et al. [52]**	On the interaction of attentional focus and gaze: the quiet eye inhibits focus-related performance decrements	*J. Sport Exerc. Psychol*. **2014**, *36*, 392–400	2014	9	1.50	1

**Table 2 ijerph-17-07574-t002:** Number of publications by research area.

Category	Frequency	Centrality	Degree
Sports sciences	44	0.27	23
Psychology	33	0.10	16
Social sciences, other topics	29	0.07	10
Ophthalmology	28	0.08	10
Hospitality, leisure, sport, and tourism	25	0.01	8
Engineering	23	0.62	29
Computer science	23	0.18	23
Neurosciences and neurology	16	0.14	15
Psychology, multidisciplinary	13	0.01	8
Psychology, applied	13	0.00	6

**Table 3 ijerph-17-07574-t003:** Top 10 journals with the most publications.

Journal	Total Publications	Impact Factor (2019)	Quartile Score	SJR (Scimago Journal & Country Rank) (2019)	Citations/Docs (2 Years)	Total Citations (2019)	Centrality	*h-*Index	Country
*Optometry and Vision Science*	12	1.46	Q1	0.89	1.789	1011	0.00	92	United States
*Physician and Sports Medicine*	12	1.66	Q1	0.82	1.792	407	0.00	41	United Kingdom
*Frontiers in Psychology*	12	2.07	Q1	0.91	2.536	17,548	0.00	95	Switzerland
*Medicine and Science in Sports and Exercise*	10	4.03	Q1	1.89	4.053	4257	0.00	216	United States
*Eye and Contact Lens Science and Clinical Practice*	9	1.52	Q2	0.74	2.099	663	0.00	54	United States
*British Journal of Ophthalmology*	7	3.61	Q1	1.88	4.026	3591	0.00	146	United Kingdom
*Acta Ophthalmological*	7	3.36	Q1	1.42	3.304	2369	0.00	82	United States
*British Medical Journal*	7	17.21	Q1	2.05	4.235	16,584	0.00	412	United Kingdom
*Journal of Sport Exercise Psychology*	7	2.24	Q1	1.20	2.013	366	0.00	93	United States
*Clinical and Experimental Optometry*	6	1.92	Q2	0.75	2.034	559	0.00	51	United States

**Table 4 ijerph-17-07574-t004:** Top 10 authors with the largest number of publications.

Author	Number of Publications	*h-*Index	Total Citations	Citation Average	Centrality	Degree
Mann DL	10	4	76	7.6	0.00	3
Balcer LJ	6	5	157	26.17	0.00	14
Galetta SL	6	5	157	26.17	0.00	14
Vater C	6	3	25	4.17	0.00	3
Hasanaj L	4	2	87	21.75	0.00	10
Hossner EJ	4	3	22	5.50	0.00	3
Ravensbergen RHJC	4	3	10	2.50	0.00	1
Akhand O	3	1	6	2.00	0.00	10
Allen PM	3	2	7	2.33	0.00	1
Kredel R	3	2	19	6.33	0.00	3

**Table 5 ijerph-17-07574-t005:** Publication rate depending on the country.

Country	Publications (%)	Centrality	Degree	Half-life
United States	49 (20.94%)	0.36	20	1.5
England	30 (12.82%)	0.26	15	1.5
China	21 (8.97%)	0.05	8	0.5
Australia	18 (7.69%)	0.16	12	1.5
Spain	16 (6.84%)	0.10	7	0.5
Netherlands	13 (5.56%)	0.10	6	1.5
Germany	12 (5.13%)	0.00	3	0.5
Japan	11 (4.70%)	0.01	3	2.5
Brazil	8 (3.42%)	0.04	2	1.5
Switzerland	8 (3.42%)	0.03	7	2.5

**Table 6 ijerph-17-07574-t006:** The most used keywords.

Keyword	Frequency	Centrality	Degree
Sport	41	0.12	21
Performance	32	0.15	27
Attention	18	0.14	27
Vision	16	0.16	29
Eye movement	14	0.02	14
Expertise	14	0.09	30
Children	14	0.11	28
Traumatic brain injury	13	0.06	26
Skill	12	0.08	22
Impact	12	0.08	19
Concussion	12	0.08	27
Injury	11	0.03	9
Eye tracking	11	0.09	15
Epidemiology	11	0.09	23
Anxiety	11	0.10	26
Movement	10	0.12	20
Saccade	9	0.08	27
Perception	9	0.11	21
Quiet eye	8	0.03	17
Visual acuity	7	0.04	16
Adolescent	7	0.09	28
Protective eyewear	6	0.09	20
Information	6	0.04	15
Gaze behavior	6	0.04	18
Eye-tracking	6	0.08	14
Eye injury	6	0.04	15
Behavior	6	0.02	11
Vision impairment	5	0.01	11
Validity	5	0.05	14
United States	5	0.01	10

**Table 7 ijerph-17-07574-t007:** Information about citations network of the four main groups.

Main Groups	Number of Publications	Number of Citation Networks	Number of Citations, Median (Range)	Number of Publications with ≥4 Citations	Number of Publications in the 100 Most Cited Publications
Group 1	108	276	1 (0–18)	46	40
Group 2	80	203	1 (0–16)	34	30
Group 3	68	238	1 (0–28)	34	24
Group 4	28	40	0 (0–10)	0	6

**Table 8 ijerph-17-07574-t008:** Information about the oldest and most recent publications in the four main groups.

Group		Author	Title	Year	Total Citations
Group 1	Oldest	Blonstein [53]	Eye injuries in sport: with particular reference to squash rackets and badminton	1975	3
	Most recent	Toldi et al. [54]	Evaluation and management of sports-related eye injuries	2020	0
Group 2	Oldest	Sherman [55]	Overview of research information regarding vision and sports	1980	2
	Most recent	Vera et al. [56]	Basketball free-throw performance depends on the integrity of binocular vision	2020	0
Group 3	Oldest	Williams et al. [57]	Assessing cue usage in performance contexts: a comparison between eye-movement and concurrent verbal report methods	1997	3
	Most recent	Witkowski et al. [58]	Fighting left handers promote different visual perceptual strategies than right handers: the study of eye movements of foil fencers in attack and defense	2020	1
Group 4	Oldest	Galetta et al. [59]	Adding vision to concussion testing	2015	6
	Most recent	Peiffer et al. [60]	The influence of binocular vision symptoms on computerized neurocognitive testing of adolescents with concussion	2020	0

**Table 9 ijerph-17-07574-t009:** Main citation network groups from the subcluster in group 1.

Subcluster	1	2	3
**Number of publications**	40	27	16
**Number of citation links**	110	53	22
**First publication**	Blonstein et al., 1975 [53]	MacEwen et al., 1999 [62]	Vinger et al., 1983 [64]
**Most cited publication**	MacEwen et al., 1987 [36]	Kriz et al., 2012 [63]	Napier et al., 1996 [47]
**Most recent publication**	Micieli et al.; 2017 [61]	Toldi et al.,2020 [54]	Woo et al., 2006 [65]
**Main keywords**	Injuries, impact, prevention	Epidemiology, trauma, risk	Hockey, injuries, head
**Topic of discussion**	Ocular lesions associated with sport	Rates of emergency admissions for sport-related ocular lesions	Sports which present the highest risk of ocular lesion
**Conclusion**	Sport is becoming an increasingly significant cause of severe ocular lesions, and the promoted use of adequate ocular protection is considered to be of the utmost importance.	Ocular lesions associated with sport present a potential impact on the provision of services. It is fundamental that ophthalmologists, optometrists, and another healthcare professionals are aware of possible ocular morbidity in the case of sport traumas and the importance of providing advice on how to prevent said lesions.	The sports which are responsible for the highest number of lesions are baseball, ice hockey, and racquet sports. Specific criteria must be developed for protection glasses. Impact-resistant polycarbonate plastic lenses and frames offer optimum protection.

**Table 10 ijerph-17-07574-t010:** Main groups of citation network of subcluster in group 2.

Subcluster	1	2	3
**Number of publications**	29	16	15
**Number of citation links**	69	40	22
**First publication**	Sherman, 1980 [55]	Applegate et al., 1992 [67]	Stine et al., 1982 [39]
**Most cited publication**	Abernethy et al., 2001 [24]	Applegate et al., 1992 [67]	Stine et al., 1982 [39]
**Most recent publication**	Jorge et al., 2019 [66]	Vera et al., 2020 [56]	Schumacher et al., 2019 [68]
**Main keywords**	Vision training, exercise, movement	Visual acuity, visual impairment, perception	Anticipation, reaction time, strategies
**Topic of discussion**	Evaluating the efficacy of sports vision training programs	Importance of the optimal visual acuity in the field	Comparison between the visual skills of athletes and non-athletes
**Conclusion**	Visual training allows for improvements to be made in terms of the visual skills of athletes, leading to greater precision in the playing field. However, there is a great controversy as to whether this training actually helps improve the on-field performance; therefore, further scientific evidence is required.	A reduction in visual acuity does not have a significant influence on sports performance. The motor–perceptual system is capable of compensating for this.	Athletes demonstrated better visual skills than non-athletes. Likewise, they presented stereopsis and a more developed visual field.

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
