# Peer review of "Citations Network Analysis of Vision and Sport"

_ijerph, 2020, doi:10.3390/ijerph17207574_

Round 1

Reviewer 1 Report

This is a very interesting articles about the sport vision.
I was particularly astonished of the use of citation networks: I've never heard about it. Methods were well described, and also results were well supported by tables and figures.

I have only a couple of suggestions:

  • pay attention to grammar and syntax rules: 
    I.E.
    line 37: beginnings > beginning
    line 54: concluded that > concluded that
    line 97: consequently > therefore
  • line 90 and methods: I think you could cite a couple of references more about citation network to make your methods stronger!
  • line 47 - 53: I suggest to talk about vision role in blind athletes (https://www.researchgate.net/publication/335864776_Jump_and_balance_test_in_judo_athletes_with_or_without_visual_impairments) 
  • discussion section:
    First of all, re-state your results!
    Then, explain why you use citation network
    At the end, try to admit your limitations (i.e. what are the differences between your paper and a systematic review?) and to suggest future researches' direction

Try to be more coincise, to write more simply!

Reviewer 2 Report

I think this is an excellent analysis and hence, I have only some minor remarks

I would be better (for the reader) if the definitions of centrality and half-life as used in Table 5 were included in the text (and not just as a reference to the literature).

Hirsch uses his Spanish (Jorge) name in his publications (not the English version: George)

The definition of the h-index is rather vague: “indicates”. Moreover, the h-index as used in the text refers to a publication (not to a scientist) and is suspiciously always equal to 1.  Please correct (saying that the software yields a value of 1 is not a professional answer).

Is the peak in 2019 real?  By this I mean: is it influenced by an increase of journals, included in the WoS?

Finally, it would be interesting to know which sports are discussed (and how often) in connection with vision (shooting, tennis, baseball, ping pong, …?)
